# Perceived stress and associated factors among health care professionals working in the context of COVID-19 pandemic in public health institutions of southern Ethiopia 2020

Abinet Teshome[1], Mulugeta Shegaze[2], Mustefa Glagn[2], Asmare Getie[3]*,
Beemnet Tekabe[2], Dinkalem Getahun[3], Tesfaye Kanko[1], Tamiru Getachew[4],
Nuhamin Yenesew[5], Simeon Meskele[4], Kabtamu Tolosie[6], Zebene Temtem[5],
Teshome Yirgu[7]

1 Department of Biomedical Science, Arba Minch University, Arba Minch, Ethiopia, 2 School of Public Health, Arba Minch University, Arba Minch, Ethiopia, 3 School of Nursing, Arba Minch University, Arba Minch, Ethiopia, 4 Department of Anatomy, Arba Minch University, Arba Minch, Ethiopia, 5 Department of Psychology, Arba Minch University, Arba Minch, Ethiopia, 6 Department of Statistics, Arba Minch University, Arba Minch, Ethiopia, 7 Department of Geography, Arba Minch University, Arba Minch, Ethiopia

* asmaregetie2017@gmail.com

**Data Availability Statement:** All relevant data are within the paper and its Supporting Information files.

## Abstract

### Introduction

Health care professionals are at higher risk of developing stress-related problems during outbreaks, due to the overwhelming clinical workload, fear of contagion, and inadequate protective gears. So, in order to monitoring mental health issues and to understand the factors evidence-based interventions is important. Therefore, this study was aimed to assess perceived stress and associated factors among health care professionals working in the context of COVID-19, Southern Ethiopia.

### Methods

Institution based cross-sectional study was conducted among 798 health care professionals from the 1st May to 1st June 2020. The study participants were selected using simple random sampling technique after allocating a proportion to each health institute based on the size of health care professionals. A pre-tested and structured interviewer-administered questionnaire using KOBO collect survey tool was used to collect data. A total score of >20 points was considered as the cut off for experiencing perceived stress based on perceived stress scale. Both bivariable and multivariable logistic regression analysis were performed to identify associated factors. The level of statistical significance was set at a p-value of less than 0.05 in multivariable logistic regression.

### Result

Nearly two-thirds 61.8% (95% CI: 58.4%, 65.2%) of HCPs had perceived stress. Not having COVID-19 updated information (AOR = 2.41, 95% CI: 1.31, 4.43), not at all confident on

**Funding:** The author(s) received no specific funding for this work.

**Competing interests:** The authors have declared that no competing interests exist.

**Abbreviations:** HCPs, healthcare providers; CI, confidence interval; AOR, adjusted odds ratio; WHO, world health organization.

coping with stress (AOR = 9.94, 95% CI:3.74, 26.41), somewhat confident in coping with stress (AOR = 4.69, 95% CI:2.81, 7.84), moderately confident on coping with stress (AOR = 2.36, 95% CI: 1.46, 3.82), and not getting along well with people (AOR = 4.88, 95% CI: 1.42, 16.72) were positively association with perceived stress. However, feeling overwhelmed by the demand of everyday life (AOR = 0.52 95% CI: 0.35, 0.77) and worrying about what other people think about them (AOR = 0.48, 95% CI: 0.24, 0.81) were negatively associated with perceived stress.

## Conclusion

COVID-19 update, confidence in coping with stress, getting along with people, worrying about what other people think about them, and feeling overwhelmed by the demand of everyday life were factors significantly associated with perceived stress. The provision of COVID-19 update to HCPs along with wider strategies to support their psychological wellbeing is vital.

## Introduction

The COVID-19 pandemic was caused by a novel corona-virus first discovered in Wuhan, Hubei province of China in December 2019. A highly infectious serious acute respiratory syndrome caused by a novel corona-virus (SARS-CoV-2) emerged in Wuhan, China, and on March 11th 2020, the World Health Organization (WHO) declared COVID-19 as a pandemic [1, 2]. The Corona-virus belongs to large groups of viruses that cause serious health problems including the mental health of the society particularly the Health care providers'. Corona virus disease (COVID-19) is a new strain that was first discovered in 2019 in Wuhan, China and has not been previously identified in humans [3].

Corona virus pandemic fears, prompt government to activate emergency response and extend travel ban. Healthcare staff is at high risk of mental health problems due to the overwhelming clinical workload, fear of contagion, and inadequate protective gears. The healthcare professionals are at high risk of developing stress-related problems during outbreaks. A study conducted in Iraq showed that more than two-thirds of health care professionals had a moderate level of stress, nearly one-fifth had low and 9.6% of health care professionals had a high level of stress [3].

COVID-19 has a serious impact on the mental health of the public. The public has shown perceived stress behaviors, causing a significant shortage of medical masks and alcohol across the country. In addition, many medical staff work more than 16 hours a day on average, causing them to not getting enough sleep. Several studies showed that mental health problems could occur in both healthcare professionals and SARS survivors during the SARS epidemic [1–3]. A study showed that the pooled prevalence of stress among health care professionals was 34% and Pooled prevalence rate of psychological morbidities with respect to impact of event due to COVID-19 pandemic was 44%. The burden of these psychological morbidities was highest among the COVID-19 patients followed by healthcare professionals and general population [4]. In the early days of the outbreak, many medical workers in Iran became infected and died. As a result of close contact with the infected patients, the health care professionals had high-level concerns about the pandemic, uncertainty of its duration, and the possibility of transmitting the disease to their family members. The most important causes of

fatigue and stress in the healthcare professionals are wearing heavy protective equipment for a long time and difficulty in performing care procedures [4, 5].

The prevalence of psychological morbidities, in any pandemic situation tends to be higher compared to normal situations. Health professionals face problems due to increased work load, intense working schedule and increased chance of getting exposed to positive cases. All these problems have in turn increased the fear of social isolation, loneliness, fear of getting the disease and dying, staying away from family, anxiety, depression, stress, insomnia, sleep disturbances and psychological distress [2, 4]. There is a dearth of information regarding stress related to COVID-19 in Ethiopia. Therefore, this study aimed to assess perceived stress and its associated factors among healthcare professionals during COVID-19 pandemic in Southern Ethiopia.

## Methods

### Study area and period

This study was conducted in the Gamo, Gofa, Konso and South Omo Zones health facilities from the 1st May to the 1st June. In Gamo zone there are one general and four primary hospitals, 56 health centers and 302 health posts in which, there are 2570 health professionals providing health services (33).

Konso Zone has two hospitals, 13 health centers and 62 health posts in which there are 521 health professionals providing health services (34). Gofa zone has two hospitals, 25 health centers and 179 health posts in which there are 815 health professionals (35).

In South omo zone there are one general and two primary hospitals, 40 health centers, 228 health posts and there are 914 health professionals working in those health facilities (36).

### Study design

Institutional based cross-sectional study design was conducted.

### Source population

All health professionals working in Gamo, Gofa, Konso and South Omo zones.

### Study population

All health professionals working in the selected health institutions and quarantine centers during the study period.

### Inclusion and exclusion criteria

**Inclusion criteria.**   All health care professionals who were available during the study period and fulfill the inclusion criteria were included in the study.

**Exclusion criteria.**   Health care professionals who were on maternal leave, annual leave and leave due to illness were excluded from the study.

### Sample size determination and sampling procedure

**Sample size determination.**   A single population proportion formula $((Z\alpha/_2)\ pq/d^2)$ was used to estimate the sample size required for the study. The sample size calculation assumed the proportion (p), the estimated level of stress among HCPs was estimated to be 50% because there was no prior study finding in Ethiopia Previously, 95% confidence level, margin of error

of 5%, and a design effect of 2 which gave the sample size of 768. In consideration of a 10% non-response rate, the final sample size was 845.

**Sampling procedure.** The study participants were selected using a multistage sampling technique. First, twenty percent of health institutions were selected using a simple random sampling technique (computer-generated random numbers) after allocating a proportion to each Zone based on the size of health institutions. Then, the sample size was proportionally allocated to the health institutes based on the size of health care workers. Lists of active health care workers were taken from each selected health institute. Finally, a simple random sampling technique (computer-generated random number) was implemented to recruit the health professionals in each selected health institute.

## Data collection tools and methods

Quantitative data was collected using a pretested structured questionnaire using KOBO collect survey tool, which was designed by reviewing different literature and WHO guideline. The questionnaire was prepared originally in English and was translated to Amharic, and then translated back to English to ensure consistency in meaning. The tool consisted of Socio-Demographic and Medical History related (12 items), Psycho-social context (22 items), Psychological related (32 items) and 10 item Perceived Stress Scale (PSS-10) (39). The data collection instrument was pretested, translated and standardized to assure its validity and reliability. During data collection, a reliability analysis was done and the result showed a good score of internal consistency between the items (Cronbach's alpha = 0.84).

## Data collectors and data quality controls

Training was given for the data collectors and supervisors for three days. Pretest was done and amendment was performed accordingly. The data was collected using ten BSC nurses and the supervision was undertaken by five MSC health care professionals.

## Data processing and analysis

The collected data were downloaded from the KOBO collect. It was then edited and cleaned for inconsistencies, missing values using excel, and then exported to SPSS version 25 (SPSS Inc., Chicago, IL, USA) for further analysis. Descriptive statistics was used to describe the study population in relation to relevant variables.

Both bivariate and multivariate logistic regression analysis were performed to identify associated factors. The variables with $P<0.2$ in the bivariable analysis were taken as a candidate for multivariable analysis. The final model was fitted using stepwise selection methods (backward conditional). Model fitness was checked using the Hosmer and Lemeshow goodness of fitness test (P-value $\geq0.05$). Odds ratio with its 95% confidence interval was used to determine the degree of association. Level of statistical significance was set at a p-value of less than 0.05.

## Data quality control

Training was given for the data collectors and supervisors. Proper instruction on how to collect and send data was thoroughly covered in the training. Pretest was done before the actual data collection, and then the modification of the tool was done based on the result of pretest. At the end of data collection completeness, accuracy, and consistency of the questionnaire was checked.

## Operational definition

**Perceived stress.**    10 Likert scale questions measuring perceived stress status of the respondents was used. The score of stress assessing question was calculated for each respondent then overall score was computed and the status was classified in to no perceived stress and perceived stress. A total score of >20 points was considered as the cut off for experiencing perceived stress based on perceived stress scale.

## Ethics approval and consent to participate

Ethical approval was obtained from institutional research ethics review board of College of Medicine and Health Science, Arba Minch University. A formal letter was written to study zones to obtain permission to conduct the research in district and a permission letter was obtained from each health facility. Finally, informed oral consent was obtained from the study participants before undertaking the study.

## Results

### Socio-demographic characteristics of the respondents

From the total sample 798 completed the interview with response rate of 94.2%. More than half 482(60.4%) of the respondents were males and nearly two-third 492(61.75) of the respondents were married and regarding educational status 363(45.5%, and 359(45%) of the respondents were diploma and first-degree holders respectively. Regarding the profession of the respondents 356 (44.6%) of them were clinical nurse, followed by public health officer 96(12%) (**Table 1**).

### Assigned departments and medical history of health care professionals

From the total respondents nearly one-fourth 187(23.4%) of the respondents were working in outpatient department (OPD) followed by one-fifth 160 (20.1%) in the obstetrics and gynecologic ward. Regarding to the medical problems thirteen (1.6%) of the respondents had history of asthma followed by diabetes mellitus which was six.

### Psychosocial factors and substance use of the respondent

From the total respondents more than half 409(51.3%) of health care professional's responded that the community response to COVID -19 as somewhat followed by 247(31%) not at all. Most of the health care professionals 663(83.1) were valued by their families as they are for being a member of COVID 19 response team and nearly three-fourth 576(72.2) of the respondents were valued by the community as for being a member of COVID 19 response team. Majority of the respondents 688(86.2%) had no history of contact with Confirmed or suspected cases. From the total respondents 678 (85%) of health care professional's responded that the availability of PPE in their working health institution was not adequate. Regarding the social media nearly three-fourth 569(71.3%) of the respondents had used both mainstream and social media. Majority of the respondents 693(86.8%) had received updated information about COVID 19 from different sources and 85.5% of health care professionals had feeling of susceptibility to contract the COVID 19.More than three-fourth of the respondents 633 (79.3%) had COVID-19 related worry and 630(78.9%) of the health care professionals believed that the suggested prevention and control practices can contain the pandemic of COVID 19. From the total health care professionals nearly one fifth (20.8%) drink alcohol, more than half (58%) perform daily physical exercise and only 1% smoke cigarette (**Table 2**).

**Table 1. Socio-demographic and medical history of health care professionals working in response to COVID-19 in Southern Ethiopia 2020.**

| Variable | Category | No. | % |
|---|---|---|---|
| Sex | Female | 316 | 39.6 |
| | Male | 482 | 60.4 |
| Marital Status | Single | 306 | 38.3 |
| | Married | 492 | 61.7 |
| Do you have children? | No | 353 | 44.2 |
| | Yes | 445 | 55.8 |
| Residence | Rural | 93 | 11.7 |
| | Town | 705 | 88.3 |
| Living arrangement | Alone | 239 | 29.9 |
| | Friends | 20 | 2.5 |
| | Parents | 161 | 20.2 |
| | Spouse | 378 | 47.4 |
| Profession | Clinical Nurse | 356 | 44.6 |
| | Medical Doctor | 65 | 8.1 |
| | Medical laboratory | 84 | 10.5 |
| | Midwifery | 120 | 15.0 |
| | Pharmacist | 77 | 9.6 |
| | Public health officer | 96 | 12.0 |
| Religion | Muslim | 38 | 4.8 |
| | Orthodox | 338 | 42.4 |
| | Protestant | 417 | 52.3 |
| | Other | 5 | 0.6 |
| Education Status | Diploma | 363 | 45.5 |
| | First degree | 359 | 45.0 |
| | Level 3 or 4 | 60 | 7.5 |
| | Second degree | 16 | 2.0 |
| Ward/Department | Dispensary | 61 | 7.6 |
| | Emergency ward | 98 | 12.3 |
| | Laboratory | 67 | 8.4 |
| | Gyn/Obs ward | 160 | 20.1 |
| | Medical ward | 98 | 12.3 |
| | OPD | 187 | 23.4 |
| | Pediatrics ward | 66 | 8.3 |
| | Surgical ward | 61 | 7.6 |
| Medical Conditions/Comorbidity | Anemia | 2 | 0.3 |
| | Asthma | 13 | 1.6 |
| | DM | 6 | 0.8 |
| | DM and Hypertension | 1 | 0.1 |
| | HBV | 4 | 0.5 |
| | HBV Prior and Anxiety | 1 | 0.1 |
| | Hypertension | 5 | 0.6 |
| | Nephrolitiasis | 1 | 0.1 |
| | TB | 1 | 0.1 |
| | Prior Anxiety | 1 | 0.1 |
| | Prior Stress | 3 | 0.4 |
| | PUD | 1 | 0.1 |
| | Sinusitis | 5 | 0.6 |
| | No | 754 | 94.5 |

**Table 2. Individual psychosocial factors and substance use of health care professionals working during COVID-19 pandemic in southern Ethiopia 2020.**

| Variable | Category | No. | % |
|---|---|---|---|
| Community members response to COVID-19 | Not at all | 247 | 31.0 |
| | Somewhat | 409 | 51.3 |
| | Moderately | 129 | 16.2 |
| | To a great extent | 13 | 1.6 |
| Do you have Valued by family | No | 135 | 16.9 |
| | Yes | 663 | 83.1 |
| Do you have Valued by the community | No | 222 | 27.8 |
| | Yes | 576 | 72.2 |
| Contact with Confirmed or suspected cases | No | 688 | 86.2 |
| | Yes | 110 | 13.8 |
| Availibity of PPE | No | 678 | 85.0 |
| | Yes | 120 | 15.0 |
| Type of media | Social | 76 | 9.5 |
| | Mainstream | 121 | 15.2 |
| | No | 32 | 4.0 |
| | Both | 569 | 71.3 |
| Having COVID-19 updates | No | 105 | 13.2 |
| | Yes | 693 | 86.8 |
| Government support on preventing and controlling | Not at all | 223 | 27.9 |
| | Somewhat | 410 | 51.4 |
| | Moderately | 141 | 17.7 |
| | To a great extent | 24 | 3.0 |
| Confident on coping with stresses | Not at all | 96 | 12.0 |
| | Somewhat | 261 | 32.7 |
| | Moderately | 307 | 38.5 |
| | To a great extent | 134 | 16.8 |
| Feeling of susceptiblity | No | 116 | 14.5 |
| | Yes | 682 | 85.5 |
| COVID-19 related worry | No | 165 | 20.7 |
| | Yes | 633 | 79.3 |
| Believe the suggested prevention and control practices control the pandemic | No | 168 | 21.1 |
| | Yes | 630 | 78.9 |
| Drink alcohol | No | 632 | 79.2 |
| | Yes | 166 | 20.8 |
| Smoke Cigarette | No | 790 | 99.0 |
| | Yes | 8 | 1.0 |
| Physical exercise | No | 335 | 42.0 |
| | Yes | 463 | 58.0 |
| Having anyone you can trust and confide in | No | 124 | 15.5 |
| | Yes | 674 | 84.5 |
| Get along well with people | No | 54 | 6.8 |
| | Yes | 744 | 93.2 |
| Feel overwhelmed by the demands of everyday life | No | 439 | 55.0 |
| | Yes | 359 | 45.0 |
| Feeling of dying due to COVID-19 | No | 572 | 71.7 |
| | Yes | 226 | 28.3 |

(*Continued*)

**Table 2.** (Continued)

| Variable | Category | No. | % |
|---|---|---|---|
| Influenced by people with strong opinions | No | 552 | 69.2 |
| | Yes | 246 | 30.8 |
| Worry about what other people think of you | No | 649 | 81.3 |
| | Yes | 149 | 18.7 |

## Perceived stress

Out of the total study participants, nearly two-thirds 61.8% (95% CI: 58.4%, 65.2%) of HCWs had perceived stress (**Fig 1**).

## Factors associated with perceived stress

The binary logistic regression analysis showed that working department/ward, having COVID-19 updates, government support on preventing and controlling, confident in coping mechanisms with stresses, get along well with people, feel overwhelmed by the demands of everyday life, and Worry about what other people think about them were found to be significantly associated with perceived stress.

The odds of perceived stress among health care workers who didn't get COVID updated information were nearly two times more likely to have perceived stress than who got an update (AOR = 2.41, 95% CI: 1.31, 4.43). When compared to their counter parts, the odds of perceived stress was higher among HCPs who were not at all confident on coping with stress (AOR = 9.94, 95% CI: 3.74, 26.41), somewhat confident in coping with stress (AOR = 4.69, 95% CI: 2.81, 7.84), and moderately confident in coping with stress (AOR = 2.36, 95% CI: 1.46, 3.82).

Similarly, the odds of perceived stress among HCPs who were not getting along well with people were five times more likely to develop perceived stress than their counterparts (AOR = 4.88, 95% CI: 1.42, 16.72) (**Table 3**).

## Discussion

Outbreaks of novel COVID-19 can be extremely stressful and detrimental to the wellbeing of health care workers (HCPs) [6]. It is crucial to recognize the health care workers that have

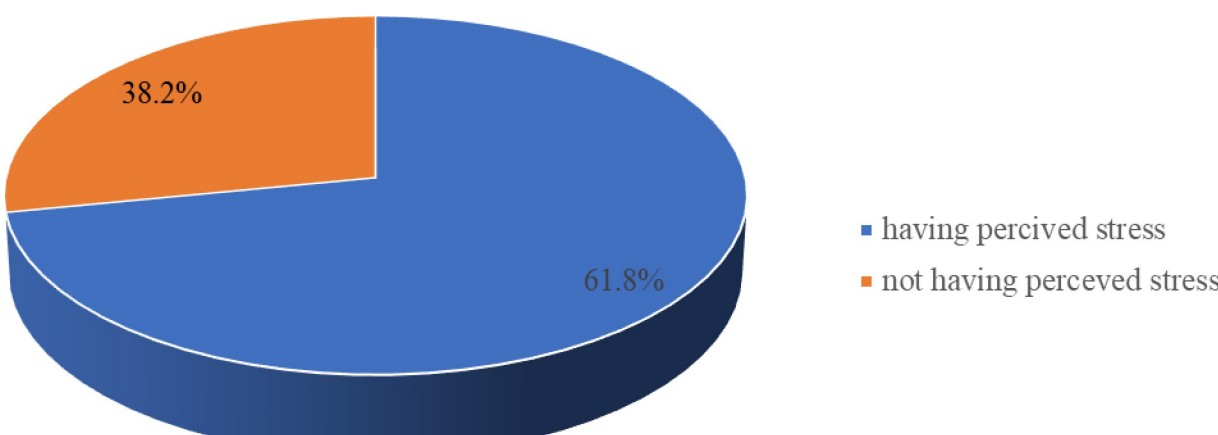

**Fig 1. Perceived stress level of health care workers who are working with COVID-19 in public health institutions of southern Ethiopia 2020.**

**Table 3. Result of binary analysis of health care professionals working in response to COVID-19 in southern Ethiopia 2020.**

| Variables | Perceived stress | | p-value | AOR with 95% CI |
|---|---|---|---|---|
| | Moderate PS N (%) | Low PS N (%) | | |
| **Residence** | | | | |
| Town | 423(85.8%) | 282(92.5%) | .421 | 0.77 (0.41–1.44) |
| Rural | 70 (14.2%) | 23 (7.5%) | | 1 |
| **Living status** | | | .648 | |
| Alone | 164(33.3%) | 75(24.6%) | .279 | 1.27 (0.82–1.99) |
| Friends | 12(2.4%) | 8(2.6%) | .666 | 0.78 (0.25–2.40) |
| Parents | 97(19.7%) | 64 (21.8%) | .827 | 1.05 (0.66–1.68) |
| Spouse | 220 (44.6%) | 158(51.8%) | | 1 |
| **Education level** | | | .209 | |
| Diploma | 245(49.7%) | 158(38.7%) | .792 | 1.21(0.29–4.98) |
| First degree | 209(42.4%) | 150(49.2%) | .947 | 1.04 (0.25–4.26) |
| Level 3 or 4 | 27(5.5%) | 33(10.8%) | .475 | 0.56 (0.12–2.67) |
| Second degree | 12(2.4%) | 4(1.3%) | | 1 |
| **Ward/Department** | | | | |
| Dispensary | 31(6.3%) | 30(9.8%) | .310 | 0.636(0.26–1.52) |
| Emergency | 67(13.6%) | 31(10.2%) | .390 | 1.41 (0.64–3.12) |
| Laboratory | 44(8.9%) | 23(7.5%) | .116 | 1.98 (0.84–4.68) |
| Gyn/Obs | 97(19.7%) | 63(20.7%) | .579 | 1.22 (0.59–2.52) |
| Medical | 76(15.4%) | 22(7.2%) | .007 | 3.07 (1.36–6.92) |
| OPD | 118(23.9%) | 69(22.6%) | .834 | 1.07 (0.53–2.18) |
| Pediatrics | 28(5.7%) | 38(12.5%) | .312 | 0.64(0.27–1.50) |
| Surgical (including OR) | 32(6.5%) | 29(9.5%) | .312 | 1 |
| **response of community members** | | | .435 | |
| Not at all | 89(18.1%) | 7(2.3%) | .345 | 1.96 (0.48–8.03) |
| Somewhat | 195(39.6%) | 66(21.6%) | .616 | 1.45(0.35–5.69) |
| Moderately | 160(32.5%) | 147(48.2%) | .645 | 1.39 (0.33–5.73) |
| To a great extent | 49(9.9%) | 85(27.9%0 | | 1 |
| **Valued by family** | | | | |
| No | 92(18.7%) | 43(14.1%) | .345 | 0.76 (0.44–1.32) |
| Yes | 401(60.5%) | 262(85.9%) | | 1 |
| **Valued by the community** | | | | |
| No | 158(32%) | 64(21%) | .382 | 1.22 (0.77–1.92) |
| Yes | 335(68%) | 241(79%) | | 1 |
| **Availibity of PPE** | | | | |
| No | 398(80.7%) | 280(91.8%) | .144 | 0.61 (0.32–1.17) |
| Yes | 95(19.3%) | 25(8.2%) | .210 | 1 |
| **Type of Media** | | | | |
| Social | 55(11.2%) | 21(6.9%) | .216 | 1.526(0.78–2.97) |
| Mainstream | 77(15.6%) | 44(14.4%) | .536 | 1.17 (0.71–1.92) |
| No | 19(3.9%) | 13(4.3%) | .143 | 0.46 (0.16–1.29) |
| Both | 342(69.4%) | 227(74.4%) | | 1 |
| **COVID-19 updates** | | | | |
| No | 76(15.4%) | 29(9.5%) | **.005** | 2.41 (1.312–4.430) |
| Yes | 417(84.6%) | 276(90.5%) | | 1 |
| **Government support** | | | **.009** | |
| Not at all | 174(35.3%) | 49(16.1%) | .098 | 2.47 (0.84–7.21) |

*(Continued)*

**Table 3.** (Continued)

| Variables | Perceived stress | | p-value | AOR with 95% CI |
|---|---|---|---|---|
| | Moderate PS N (%) | Low PS N (%) | | |
| Somewhat | 231(46.9%) | 179(58.7%) | .549 | 1.37 (0.48–3.84) |
| Moderately | 75(15.2%) | 66(21.6%) | .912 | 0.942 (0.32–2.71) |
| To a great extent | 13(2.6%) | 11(3.6%) | | 1 |
| **Confident on coping with stress** | | | **.001** | |
| Not at all | 89(18.1%) | 7(2.3%) | .001 | 9.94 (3.74–26.41) |
| Somewhat | 195(39.6%) | 66(21.6%) | .001 | 4.69 (2.80–7.83) |
| Moderately | 160(32.5%) | 147(48.2%) | .001 | 2.36 (1.45–3.82) |
| To a great extent | 49(9.9%) | 85(27.9%) | | 1 |
| **Feeling of susceptiblity** | | | | |
| No | 87(17.6%) | 29(9.5%) | .289 | 1.36 (0.77–2.40) |
| Yes | 406(82.4%) | 276(90.5%) | | 1 |
| **Physical exercise** | | | | |
| No | 187(37.9%) | 148(48.5%) | .176 | 0.78 (0.54–1.11) |
| Yes | 306(62.1%) | 157(51.5%) | | 1 |
| **Have anyone to trust and confide in** | | | | |
| No | 99(20.1%) | 25(8.2%) | .176 | 1.50 (0.83–2.71) |
| Yes | 394(79.9%) | 280(91.8%) | | 1 |
| **Get along well with people** | | | | |
| No | 50(10.1%) | 4(1.3%) | **.012** | 4.87 (1.42–16.71) |
| Yes | 443(89.9%) | 301(98.7%0 | | 1 |
| **Feel overwhelmed by the demands of everyday life** | | | | |
| No | 236(47.9%) | 203(66.6%) | **.001** | 0.52 (0.35–0.77) |
| Yes | 257(52.1%) | 102(33.4%) | | 1 |
| **Feel you cannot make it** | | | | |
| No | 333(67.5%) | 239(78.4%) | .438 | 0.84 (0.55–1.29) |
| Yes | 160(32.5%) | 66(21.6%) | | 1 |
| **Influenced by people with strong opinions** | | | | |
| No | 333(67.5%) | 219(71.8%) | .972 | 0.99 (0.66–1.47) |
| Yes | 160(32.5%) | 86(28.2%) | | 1 |
| **Worry about what other people think of you** | | | | |
| No | 380(77.1%) | 269(88.2%) | **.005** | 0.48 (0.29–0.80) |
| Yes | 113(22.9%) | 36(11.8%) | | 1 |

perceived stress to enable timely intervention. Hence, the study aimed to assess the perceived stress and its associated factors among health care workers working during COVID-19 pandemic in public health institution of southern Ethiopia. The study revealed that nearly two-thirds 61.8% (95% CI: 58.4%, 65.2%) of the respondents had perceived stress. The study finding coincides with the study finding from Ghana, which reported that 64% of health care professionals had moderate stress [7]. The result might be due to low perceived preparedness to respond to the COVID-19 pandemic among HCPs. Low perceived preparedness was associated with increased stress [7]. In addition, healthcare workers are particularly exposed to negative mental health effects during caring for patients. The ongoing COVID-19 pandemic is also taking a massive toll on the mental and emotional well-being of healthcare professionals around the world [7, 8].

However, this study finding was slightly higher than the study findings from China, 53.8% [9]. The possible explanation might be due to the fact that the current study was conducted in

the resources limited setting. Thus, health care workers are facing physical separation and increased in care demands, equipment shortages, and the higher risk of COVID-19 infection, which make the health professional prone to developing perceived stress [8, 10]. The supporting evidence on variation might be, the finding from China has been generated from the general population; however, in the current study was from health care providers.

In the current study, health care workers who didn't get COVID-19 updated were more likely to have perceived stress than their counterparts. Effective process of COVID-19 risk communication for health can improve the understanding on the cause and impact of the risk and promotes protective behaviors among health care professionals [11]. The study finding conducted among health care professionals in China showed that providing accurate information for health care professionals is decisive to alleviate their perceived threat [12]. It also argued by a literature review [13], a clear communication of directives and precautionary measures are used to reduce stress. This can be explained by rapidly evolving nature of pandemics and the frequency with which information changes, HCPs who did not know where to go for up-to-date guidance are prone to stress [6].

The study finding indicated that respondents who were somewhat confident on coping with stress were nearly five times more likely to develop perceived stress than those respondents who had a great confident. Similarly, health care professionals who are moderately confident on coping with stress were nearly three times more likely to have perceived stress than those who had a great confidence. The finding is in line with a study in Hong Kong which demonstrated that elderly individuals who had greater confidence were more likely to adopt authority-suggested preventive behaviors during the severe acute respiratory syndrome pandemic [14]. This may be explained by individual's beliefs in the ability to do specific tasks have a positive effect on prevention or coping with a problem [15].

The current study finding identified that health care professionals who were not getting along well with people were more likely to develop perceived stress than the ones getting along well with people. A similar finding, indicating that getting along well with people may decrease perceived stress in the pandemic and increase confidence in coping with COVID-19 [16]. However, the familiar ways to interact is interrupted during a pandemic [6, 17]. Due to that the health care professionals may feel incapable to cope with stress and resulting to develop perceived stress.

In these study findings the age and sex of health care professionals were not determinants of perceived stress. The finding is inconsistent with the study finding from Ethiopia, in which females were 2.39 times more likely to experience high perceived stress than men [18]. The variation might be due to the sample size, design and population.

This study had limitations that must be considered when taking the data. The results represent a point in time at an early stage of preparations in the study area. Only public health institution in four zones in Ethiopia surveyed mean that the results may not be generalizable across a nation. In addition, as being cross-sectional in the design, it does not confirm the definitive cause and effect relationship. In addition, the study did not assess the stress in health care professionals working in separate COVID-19 treatment centers.

## Conclusions

In conclusion, this study discovered that around two third of healthcare providers were found to have perceived stress relating to COVID-19. Lack of line of commutation to providing up to date information, lack of confidence on coping stress and not getting along well with individuals are factors associated with developing perceived stress. Therefore, precautionary measures such as making sure of availability of a well- established line of communication for the

healthcare providers can have a great role in preventing and alleviating the perceived stress and safeguard the professionals. In addition, providing psychological support to develop confidence on coping strategies and getting along with fellow professionals and others can have a positive impact in relation to stress.

## Supporting information

**S1 File. Data collection tool.**
(DOCX)

**S2 File. The dataset used for this study.**
(SAV)

## Author Contributions

**Conceptualization:** Abinet Teshome, Mustefa Glagn, Asmare Getie, Beemnet Tekabe, Tesfaye Kanko, Tamiru Getachew, Nuhamin Yenesew, Simeon Meskele, Kabtamu Tolosie, Zebene Temtem, Teshome Yirgu.

**Formal analysis:** Abinet Teshome, Mulugeta Shegaze, Mustefa Glagn, Asmare Getie, Tesfaye Kanko, Nuhamin Yenesew, Kabtamu Tolosie, Teshome Yirgu.

**Investigation:** Abinet Teshome, Mulugeta Shegaze, Mustefa Glagn, Asmare Getie, Tesfaye Kanko.

**Methodology:** Abinet Teshome, Mulugeta Shegaze, Mustefa Glagn, Asmare Getie, Beemnet Tekabe, Dinkalem Getahun, Tesfaye Kanko, Nuhamin Yenesew, Simeon Meskele, Kabtamu Tolosie, Zebene Temtem, Teshome Yirgu.

**Software:** Abinet Teshome, Mulugeta Shegaze, Mustefa Glagn, Asmare Getie, Beemnet Tekabe, Dinkalem Getahun, Tesfaye Kanko, Nuhamin Yenesew, Simeon Meskele, Kabtamu Tolosie, Zebene Temtem, Teshome Yirgu.

**Supervision:** Abinet Teshome, Mulugeta Shegaze, Mustefa Glagn, Asmare Getie, Dinkalem Getahun, Nuhamin Yenesew, Simeon Meskele, Kabtamu Tolosie, Zebene Temtem, Teshome Yirgu.

**Validation:** Abinet Teshome, Mulugeta Shegaze, Mustefa Glagn, Asmare Getie, Dinkalem Getahun, Simeon Meskele, Kabtamu Tolosie, Zebene Temtem, Teshome Yirgu.

**Writing – original draft:** Mustefa Glagn, Asmare Getie, Beemnet Tekabe, Tesfaye Kanko, Tamiru Getachew, Nuhamin Yenesew, Teshome Yirgu.

**Writing – review & editing:** Mulugeta Shegaze, Mustefa Glagn, Asmare Getie, Beemnet Tekabe, Dinkalem Getahun, Tamiru Getachew, Teshome Yirgu.

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
