## [Decision Letter · Decision Letter 0]

9 Mar 2021

PONE-D-20-38452

Perceived stress and associated factors among health care professionals working during COVID-19 pandemic in public health institutions of southern Ethiopia 2020

PLOS ONE

Dear Dr. Getie,

Thank you for submitting your manuscript to PLOS ONE. After careful consideration, we feel that it has merit but does not fully meet PLOS ONE’s publication criteria as it currently stands. Therefore, we invite you to submit a revised version of the manuscript that addresses the points raised during the review process.

Your outcome variable, PSS-10, does not have cut-off values. We expect the statistical analysis to be revised to one appropriate for continuous outcome variable. We also recommend that the PSS-10 distribution to be presented in graphs showing percentages of participants with specific scores if possible.

We look forward to receiving your revised manuscript.

Kind regards,

Markos Tesfaye, M.D., Ph.D

Academic Editor

PLOS ONE

Journal Requirements:

https://onlinelibrary.wiley.com/doi/10.1111/pcn.13128

https://www.dovepress.com/front_end/preparedness-and-responses-of-healthcare-providers-to-combat-the-sprea-peer-reviewed-fulltext-article-ID

In your revision ensure you cite all your sources (including your own works), and quote or rephrase any duplicated text outside the methods section. Further consideration is dependent on these concerns being addressed.

Additional Editor Comments:

The manuscript reports findings of a cross-sectional survey of perceived stress among health care providers working in southern Ethiopia. The authors have used PSS-10 to measure perceived stress and performed logistic regression to identify factors associated with perceived stress. While the area of research is relevant to the challenges faced during the COVID pandemic, there are issues that limit the reports scientific value.

1. The authors did not provide any cut-off value for perceived stress in their operational definitions making the interpretation of the data very difficult. It is not helpful to present a percentage of perceived stress when there are no established cut-offs.

2. Since perceived stress may not have established cut-off values, it is better to present the distribution of perceived stress in a histogram. Also, it would be more appropriate to present linear regression analysis with perceived stress as continuous outcome variable than as categorical one.

Reviewers' comments:

Reviewer's Responses to Questions

**Comments to the Author**

1. Is the manuscript technically sound, and do the data support the conclusions?

Reviewer #1: No

Reviewer #2: Partly

2. Has the statistical analysis been performed appropriately and rigorously? 

Reviewer #1: No

Reviewer #2: Yes

3. Have the authors made all data underlying the findings in their manuscript fully available?

Reviewer #1: Yes

Reviewer #2: Yes

4. Is the manuscript presented in an intelligible fashion and written in standard English?

Reviewer #1: No

Reviewer #2: Yes

5. Review Comments to the Author

Reviewer #1: Title:

1. The study is about perceived stress of healthcare workers in the context of COVID-19. May the authors include “in the context of…” instead of “during” in the title?

Abstract

2. There are editorial problems.

3. Why written consent was not obtained from the participants?

4. In the methods section of the abstract, it says: “The study participants were selected using simple random sampling technique after allocating a proportion to each health institute based on the size of HCPS. A pretested and structured interviewer-administered questionnaire using KOBO collect survey tool was used to collect data”. What are HCPS and KOBO? This should be fully spelled in first use.

5. The tools of data collection, at least for main outcomes, are not briefly stated in the abstract. May the authors also state the cut off of dichotomizing the outcome measures in this abstract section?

6. In the conclusion section of the abstract, the authors have repeated the findings than showing the readers about the implications of the findings. Indeed, the first couple of sentences look non-relevant to the conclusion.

Introduction

On page 4, line 83-84, it says: “There is a dearth of information regarding stress related to COVID-

7. 19 in Ethiopia”. What about similar study conducted in Dilla, southern Ethiopia? what about green literature in this area in Ethiopia?

Methods

8. The authors used simple random sampling. There is also a statement about multistage sampling? If multistage sampling was used, was it stratified by the types of institutions: hospital, HC or health posts?

9. KOBO” requires further description. The outcome measures and their psychometric properties including the issues of validity in the Ethiopian context is not given. Please, do it. The cut off values in the original tools and in the current manuscript are not given. Otherwise it is difficult to interpret the findings and there is no gauge.

10. With respect to the data quality, how the data was entered. Would you describe it?

Results

11. The Ethiopian population is more than 80% rural. But, the samples indicated that only 12% rural participants. This indicates non-random sampling??

12. Many sentences are not clear in the results. ex “From the total respondents more than half 409(51.3%) of health care professional’s responded that the community response to COVID -19 as somewhat followed by 247(31%) not at all”.

13. I could not follow what Table 2 is presenting? The use of language and given options for the items are not clear.

14. The following variables are not clear

a. Government support: government support is similar for similar healthcare institutions. so why it is included as variable? Moreover, how it was measured?

b. Valued by community: what does it mean? How was it assessed?

c. Response of community members: what is it? How has it been assessed?

d. What was the assessment tool for “confidence in coping”?

Conclusion

“Lack of line of commutation to providing up to date information, lack of confidence on coping stress and not getting along well with individuals are factors associated with developing perceived stress”. What does it mean?

Reviewer #2: Exclusion criteria needs to be clarified further

Who were your data collectors with their level of educational background? ( Numner of data collectors and and supervisors should be clarified)

For how many days training was given for data collectors?

Please rewrite your operational definitions

In the category of professions why you omitted professionals like mental health and environmental health workers?

6. PLOS authors have the option to publish the peer review history of their article (what does this mean?). If published, this will include your full peer review and any attached files.

Reviewer #1: No

Reviewer #2: No

---

## [Author Response · Author response to Decision Letter 0]

8 Apr 2021

Author’s Point-by-Point Response to the Reviewer's and Editors Reports

Title: Perceived stress and associated factors among health care professionals working during COVID-19 pandemic in public health institutions of southern Ethiopia 2020

Corresponding author: Asmare Getie/ asmaregetie2017@gmail.com

Authors:

Asmare Getie

ID: - PONE-D-20-38452

 Journal: PLOS ONE 

Point by point response to Reviewers and Editors 

First of all, the authors would like to thank PLOSE ONE Journal editors and the respective reviewers for reviewing our manuscript and providing the necessary comments to be corrected. As per the comments given, we have made corrections point by point to comment. The authors tried to answer all the issues raised by editorial team and reviewers. Please note that we gave our response in blue font color.

Point by point response for Editors

Question 1: The authors did not provide any cut-off value for perceived stress in their operational definitions making the interpretation of the data very difficult. It is not helpful to present a percentage of perceived stress when there are no established cut-offs.

Response 1: Thank you for your nice and constructive suggestions: we have used a total score of >20 points was considered as the cut off for experiencing perceived stress based on perceived stress scale.

Point by point response for Reviewers

Question 1: The study is about perceived stress of healthcare workers in the context of COVID-19. May the authors include “in the context of…” instead of “during” in the title?

Response 1: Thank you for your constructive comment. We have corrected accordingly as, Perceived stress and associated factors among health care professionals working in the context of COVID-19 pandemic in public health institutions of southern Ethiopia 2020

Question 2: There are editorial problems.

Response 2: thank you, the editorial problems were corrected, you can appreciate it in the track change and clean manuscript. 

Question 3: Why written consent was not obtained from the participants?

Response 3: during the data collection the transmission of COVID-19 and the fear related with COVID-19 transmission was high. And the main route of transmission was by body contact and sharing of equipment’s. to Avoid the risk of transmission the data were collected using KOBO collector software, and the researchers decided to use verbal consent. 

Question 4: In the methods section of the abstract, it says: “The study participants were selected using simple random sampling technique after allocating a proportion to each health institute based on the size of HCPS. A pretested and structured interviewer-administered questionnaire using KOBO collect survey tool was used to collect data”. What are HCPS and KOBO? This should be fully spelled in first use.

Response 4: Thank you for your nice suggestion, HCPS mean that health care professionals and it was written based on your suggestion, and KOBO collection survey is based on the open source Collect app by get ODK and is used for primary data collection in humanitarian emergencies and other challenging field environments. With this app you enter data from interviews or other primary data -- online or offline. There are no limits on the number of forms, questions, or submissions (including photos and other media) that can be saved on your device

Question 5: The tools of data collection, at least for main outcomes, are not briefly stated in the abstract. May the authors also state the cut off of dichotomizing the outcome measures in this abstract section?

Response 5: thank you for this nice comment and suggestion, the cutoff point was stated in the abstract part as well as in the operational definitions.

Question 6: In the conclusion section of the abstract, the authors have repeated the findings than showing the readers about the implications of the findings. Indeed, the first couple of sentences look non-relevant to the conclusion.

Response 6: thank you, it was amended as per recommendation. The irrelevant sentence was omitted. 

Question 7: Introduction on page 4, line 83-84, it says: “There is a dearth of information regarding stress related to COVID-19 in Ethiopia”. What about similar study conducted in Dilla, southern Ethiopia? What about green literature in this area in Ethiopia?

Response 7: a similar study that was conducted in Dilla southern Ethiopia was not was not addressed different health institutions and all relevant data cannot be founded there. Therefore scarcity of data regarding COVID-19 in the southern region was critical.

Question 8: The authors used simple random sampling. There is also a statement about multistage sampling? If multistage sampling was used, was it stratified by the types of institutions: hospital, HC or health posts?

Response 8: stratification was not used. Just we had proportionally allocated to each health institution based on the number of health care professionals and we have used simple random sampling technique to select study subjects from the selected health institutions.

Question 9: KOBO” requires further description. The outcome measures and their psychometric properties including the issues of validity in the Ethiopian context is not given. Please, do it. The cut off values in the original tools and in the current manuscript are not given. Otherwise it is difficult to interpret the findings and there is no gauge.

Response 9: KOBO collection survey is based on the open source Collect app by get ODK and is used for primary data collection in humanitarian emergencies and other challenging field environments. With this app you enter data from interviews or other primary data -- online or offline. There are no limits on the number of forms, questions, or submissions (including photos and other media) that can be saved on your device.

The outcome measures were validated in Ethiopian contexts, different studies in Ethiopia have used this tool to measure perceived stress, so the issue of validity was not a concern. The cutoff value was stated in operational definition, those who scored greater than 20 from the total 40 score were considered as they had perceived stress.

Question 10: With respect to the data quality, how the data was entered. Would you describe it?

Response 10: The data were collected using KOBO collection survey which was uploaded on smart phone, after the data was collected it was sent to the server and which was presented as EXCEL sheet. Finally the data was exported from EXCEL sheet in to SPSS version 25 for processing and analysis. 

Question 11: The Ethiopian population is more than 80% rural. But, the samples indicated that only 12% rural participants. This indicates non-random sampling??

Response 11: the study was conducted on health care professionals who were working in different health institutions and most of health institutions were available in the town starting from woreda level, because electric city is a minimum requirement for health institutions and in the rural environment the access was almost null. Therefore even if some of the study participants were born and grow in the rural community, currently they are working in health institution that were available at woreda level, special woreda and zonal level. So the respondents might be reported their residence as urban, since currently they are living there.

Question 12: Many sentences are not clear in the results. ex “From the total respondents more than half 409(51.3%) of health care professional’s responded that the community response to COVID -19 as somewhat followed by 247(31%) not at all”.

Response 12: Thank you for your nice comment, community response to COVID -19 (to what extent the community was given a response to COVID-19) is one variable which was assessed. There were four options for this variables, to a great extent, moderately, somewhat and not all. In the result part nearly more than half were responded as the community response to COVID-19 was somewhat and followed by 247 (31%) not at all.

Question 13: I could not follow what Table 2 is presenting? The use of language and given options for the items are not clear.

Response 13: we had made some modification on the language, you can appreciate it from the manuscript, and the options were used from validated and contextualized tools that were used in other previous studies.

Question14: The following variables are not clear

a. Government support: government support is similar for similar healthcare institutions. So why it is included as variable? Moreover, how it was measured?

b. Valued by community: what does it mean? How was it assessed?

c. Response of community members: what is it? How has it been assessed?

d. What was the assessment tool for “confidence in coping”?

Response 14: Government support might be similar for similar healthcare institutions, but it can be a variable, the intention of the researchers were to know the association of having or not having government support to healthcare institutions and perceived stress on health care professionals. Since it is a pandemic government support might have a significant effect on perceived stress of health care professionals.

Valued by community mean that, (does the community gave a value or recognition to your effort and work?). In the context of COVID-19 health care professionals were the frontline workers who were fighting against a pandemic to save the community and the community might have a value to health care professionals. The intention of the researchers were to know it the magnitude and its association with perceived stress. It was assessed by asking the respondents since they were living in the community, they can appreciated how the community gave value to their effort and respond it accordingly. Just by asking does the community give value for your effort to fight against COVID-19?

Response of community members mean that how the community give response to prevention and transmission of COVID 19? And there were four options: to a great extent, moderately, somewhat and not at all.

Confidence in coping stress was assessed by asking the respondents how much you are confident enough to coping stress. And it had four options: to a great extent, moderately, somewhat and not at all.

Question 15: Conclusion

“Lack of line of commutation to providing up to date information, lack of confidence on coping stress and not getting along well with individuals are factors associated with developing perceived stress”. What does it mean?

Response 15: lack of line of communication to providing up-to-date information about COVID-19, mean that lack of updated information regarding COVID-19, it was stastically significant to perceived stress. Lack of confidence in coping stress mean that the respondents were not confident to cope stressful situations and any pandemic can create a stress full situation and there was significantly association between lack of confidence in coping stress and perceived stress.

Response to reviewer 2

Question 1: Exclusion criteria needs to be clarified further

who were your data collectors with their level of educational background? (Number of data collectors and supervisors should be clarified)

For how many days training was given for data collectors?

Please rewrite your operational definitions

In the category of professions why you omitted professionals like mental health and environmental health workers?

Response 1: thank you very much for your nice suggestion, the exclusion criteria was amended accordingly in the manuscript, data collectors and supervisors including their level of educational back ground, duration of trainings were incorporated in the manuscript.

Operational definitions was rewrite again

Majority of health institutions had no mental health and environmental health workers that is why the category omitted this professionals.

---

## [Decision Letter · Decision Letter 1]

24 May 2021

Perceived stress and associated factors among health care professionals working during COVID-19 pandemic in public health institutions of southern Ethiopia 2020

PONE-D-20-38452R1

Dear Dr. Getie,

We’re pleased to inform you that your manuscript has been judged scientifically suitable for publication and will be formally accepted for publication once it meets all outstanding technical requirements.

Kind regards,

Jianguo Wang, PhD

Academic Editor

PLOS ONE

Additional Editor Comments (optional):

I checked the responses and read through the revised manuscript. It can be acceptable technically.

Reviewers' comments:

Reviewer's Responses to Questions

**Comments to the Author**

1. If the authors have adequately addressed your comments raised in a previous round of review and you feel that this manuscript is now acceptable for publication, you may indicate that here to bypass the “Comments to the Author” section, enter your conflict of interest statement in the “Confidential to Editor” section, and submit your "Accept" recommendation.

Reviewer #2: All comments have been addressed

2. Is the manuscript technically sound, and do the data support the conclusions?

Reviewer #2: Yes

3. Has the statistical analysis been performed appropriately and rigorously? 

Reviewer #2: Yes

4. Have the authors made all data underlying the findings in their manuscript fully available?

Reviewer #2: Yes

5. Is the manuscript presented in an intelligible fashion and written in standard English?

Reviewer #2: Yes

6. Review Comments to the Author

Reviewer #2: the author has addressed all comments, the manuscript is technically sound and all the data supports the conclusion.

7. PLOS authors have the option to publish the peer review history of their article (what does this mean?). If published, this will include your full peer review and any attached files.

Reviewer #2: No

---

## [Editor Report · Acceptance letter]

3 Jun 2021

PONE-D-20-38452R1 

Perceived stress and associated factors among health care professionals working in the context of COVID-19 pandemic in public health institutions of southern Ethiopia 2020 

Dear Dr. Getie:

I'm pleased to inform you that your manuscript has been deemed suitable for publication in PLOS ONE. Congratulations! Your manuscript is now with our production department. 

Kind regards, 

on behalf of

Dr. Jianguo Wang 

Academic Editor

PLOS ONE